# Loss2Net: Loss Meta-Learning for Regression with A-priori Unknown Metrics

## Abstract

There exist many practical applications where regression tasks must cope with a generally overseen problem: the output variable to be computed, which is often a decision variable, impacts the performance metric to optimize in a manner that is not known a priori. This challenge translates into a loss-metric mismatch, which makes standard loss functions such as Mean Square Error (MSE) not suitable because they significantly hinder the final performance. While this problem is of crucial importance in, e.g., many engineering and economic applications, the literature in meta-learning of loss functions has focused on other problems, such as classification or few-shot learning tasks. In this work, we aim at closing this research gap by proposing a model that can handle common situations in real systems where the *unknown* prediction-metric relationship is time-correlated, non-differentiable, or depends on multiple intertwined predictions. We present a novel loss meta-learning architecture for regression, named `Loss2Net`, which is able to (i) jointly learn the actual regressor and the loss function that it should minimize, directly from system responses; (ii) it does so without any assumption on the loss function structure; (iii) it provides a manner to learn non-differentiable and multi-dimensional loss functions from entangled performance metrics. Detailed experiments for power grid and telecommunications infrastructure optimization, grounded on real-world measurement data, demonstrate how `Loss2Net` can effectively learn unidentified loss functions.

## 1 Introduction

Loss functions drive the training process of supervised machine learning models. In the vast majority of cases, loss functions are designed *generic* enough to work well in a wide range of scenarios. All these loss functions have in common that their objective is to minimize a symmetric error between output and ground-truth. For example, in regression problems, Mean Absolute Error (MAE), MSE, or Mean Squared Logarithmic Error (MSLE) are common choices for expressing the loss.

However, there exist practical cases where such traditional losses do not characterize well the target performance metric of the regression task. This problem appears, for instance, in many dimensioning problems encountered in systems management for business or engineering tasks. In these problems, the goal is deciding on a system capacity that is *sufficient* to accommodate the future demand: underprovisioning of capacity leads to the disruption of the offered service, while overprovisioning just causes squandering of resources; in such use cases, it is critical that the regressor or predictor learns to output a minimum quantity that is always *above* the demand.

An additional difficulty in practical situations is that the exact relationship between the variable to be predicted and the actual performance metric is often not known a priori. For instance, let us consider the problem of the proactive allocation of assets in a mobile network infrastructure to run a video streaming service (Barakabitze et al., 2020). The mobile operator must reserve network resources to maximize the user's future Quality of Experience (QoE). Yet, how the resource allocation translates into the final per-user QoE is not known to the operator in advance: it is instead an involved relation that depends on specific service properties and the multi-domain network configuration (Mangla et al., 2018). Hence, the performance of the decision can be only observed a posteriori.

In such settings, using a fixed generic loss function that does not align with the task objective creates a loss-metric mismatch (Huang et al., 2019), while manually designing a dedicated loss function is

not feasible, since the way the prediction affects the metric is unknown at the time of model design. Instead, we argue that these practical applications call for models that are capable of *learning* the correct loss from experience, and use it to properly train the regressor.

Therefore, in this paper we investigate the problem of training a neural network (NN) model to perform regression under *unknown*, complex prediction-metric relationships, by leveraging a loss function meta-learning approach. As extensively discussed in Section 2, no solution in the existing literature tackles the practical problem above. To close this research gap, we propose `Loss2Net`,[1]a regression model that builds on a joint co-training of a main regressor network and a loss-shaper network, and takes advantage of controlled noise during training to implement the exploration of the correct loss function behavior for rarely observed samples. We later apply `Loss2Net` to representative problems in real-world engineering applications. The results demonstrate how `Loss2Net` can learn differentiable approximations of time-correlated and non-differentiable performance metrics that are *not known priori* to model deployment. By doing so, `Loss2Net` learns how to drive the regressor towards estimates that are best suited to each prediction goal without human intervention in the loss design. Overall, our study unveils the advantages of loss function meta-learning for regression, paving the way for the adoption of this paradigm in practical domains.

## 2 Related work, Challenges and Contributions

Meta-learning, which is also referred to as learning-to-learn, allows us to automatically tune several features of the learning algorithm to adapt it to the target task (Hospedales et al., 2022). Among other topics, meta-learning has proven successful in (i) acting upon training data (e.g., distillation (Wang et al., 2020), batching (Fan et al., 2018), or augmentation (Cubuk et al., 2019)); (ii) shaping model parameters (e.g., initialization (Finn et al., 2017), optimization (Andrychowicz et al., 2016), or fine-tuning (Fang et al., 2024)); and (iii) other aspects such as tuning of hyper-parameters (Micaelli & Storkey, 2021), incorporation of extrinsic features (Fang et al., 2021), handling heterogeneous scenarios (Jiang et al., 2023), discovery of the actual architecture (Liu et al., 2019), providing modular compositions (Alet et al., 2019) or few-shot object detection (Demirel et al., 2023).

Our focus is on meta-learning of loss functions, which aims at learning the parameters, components, or shape of the loss to be used to train the actual model. The problem can be seen as an instance of a hierarchical optimization, where a meta-model is optimized under a constraint represented by the main model optimization (Franceschi et al., 2018). We stress that this is semantically different from meta-learning optimization schedules in iterative and alternate optimization processes (Xu et al., 2019). In the following, we present the main related works and challenges in Section 2.1, and summarize our main contributions in Section 2.2.

### 2.1 Current State of the Art

There are three main approaches to handle the loss-metric mismatch and the need for particular loss functions: $a$) learning to parametrize a predefined loss, $b$) learning surrogate loss functions, and $c$) learning to teach. We next describe these approaches.

**a) Learning to parametrize a predefined loss.** Most works on loss meta-learning propose dedicated models to infer the most suitable configuration of a predefined, parametrizable loss function. Some studies have investigated the use of decision networks to select among a set of predefined (family of) loss functions (Denevi et al., 2018), to learn the parameters of known and differentiable meta-losses (Bechtle et al., 2021), or to adapt standard loss functions (Maldonado et al., 2023). Other relevant investigations have focused on multi-part loss functions, where the goal is setting (Huang et al., 2019) and possibly dynamically updating (Heydari et al., 2019) the function weights based on (live) performance metrics. Related to the same concept are strategies such as training a network to correct the optimization trajectory produced by a fixed loss (Huang et al., 2021), or relying on generic loss functions containing hyper-parameters to be learned during training along with the neural network parameters (Barron, 2019). In all these cases, the loss — independently of whether it is expressed as a tunable function or a set of primitives — must be designed or selected manually, which is not possible when the performance metric of interest is not known a priori.

---

[1]We will make the code publicly available upon publication to ensure the reproducibility of our results.

**Learning surrogate loss functions.** *Surrogate losses* are often used as a proxy for discontinuous or otherwise challenging prediction-metric relationships that cannot be directly used to drive the learning process. Surrogates are typically task-specific and handcrafted, which involves significant manual effort for each new task — and of course requires prior knowledge about the aforementioned relationship. To remove the need for time-consuming human expert intervention, recent proposals have explored the possibility of meta-learning surrogates. Solutions in this space have proposed to express the performance metric as a function of a set of simple surrogates (Jiang et al., 2020), or to compose a loss function from primitive mathematical operators (Li et al., 2022; Raymond et al., 2023). A more elaborated strategy in this direction is that of learning the loss as a convex-by-construction function within a given parametric family (e.g., MSE or other quadratic operations of the loss input) and identifying the exact best parameters (Shah et al., 2022).

All the proposals above still require a priori knowledge of the original relationship to identify a relevant set or family of surrogates, hence do not answer to the need of learning the loss at model runtime upon deployment in the target system, which is our main goal. A closer design to the one we adopt for `Loss2Net` is a clean-slate surrogate learning based on a neural network modeling of the loss (Grabocka et al., 2019). Yet, the approach is intended for classification tasks only and is not adapted for regression: it explicitly makes the result invariant to the ordering of the minibatch samples, which is discrepant with time series forecasting, and it adopts a bilevel programming optimization of the model parameters that we later show not to perform well in regression tasks.

**Learning to teach.** The concept of representing the loss function via a neural network has been in fact explored beyond the context of surrogate losses, as the so-called *learning to teach* paradigm. In an early work, the use of a teacher network was proposed to dynamically train parameters of a loss function that adapts to the learning stage of the main model (Wu et al., 2018); yet, this approach still relies on a generic known loss function to be parametrized by the teacher. The seminal idea of a trainable task-parametrized and clean-slate loss generator was introduced for reinforcement and supervised learning by the *meta-critic* model, where an action-value function neural network learns to criticise the actions in a specified task (Sung et al., 2017). However, the meta-critic model is only applied to supervised learning problems as a tool for pre-training in the few-shot learning of new tasks (e.g., by generalizing to unseen value ranges in the same domain), and is explicitly indicated by its authors as inappropriate for single tasks like the ones we consider.

It is also worth noting that meta-critic and its extensions (Jiang et al., 2020; Gao et al., 2021) are dedicated to discrete-space classification tasks or ranking problems. The same is also true of recent proposals to employ genetic programming tools to learn clean-slate loss functions (Gonzalez & Miikkulainen, 2020; Morgan & Hougen, 2024). Instead, little attention has been paid until now to loss meta-learning for regression, and the main advances in this topic focus on multi-task learning with manually selected single-task loss functions (Raymond et al., 2023).

## 2.2 Challenges and Novelty of our contribution

Overall, prior studies on loss meta-learning have focused on (i) parametrizable loss functions or (ii) clean-slate losses for classification tasks. Little attention has been paid to the meta-learning of clean-slate losses for regression. Part of the reason comes from the fact that loss meta-learning has been considered to create an indirection that makes single-task regression less efficient (Sung et al., 2017), under the assumption that losses such as MAE or MSE can already optimally drive training in that case. Our study challenges this assumption and shows that there exist practical use cases where loss meta-learning benefits single-task predictions. Also, all literature solutions are designed for scenarios where perfect knowledge of the prediction-metric relationship is available at model design time. To the best of our knowledge, ours is the first study to propose a solution for practical use cases where a priori information about such a relationship is not available.

## 3 A Model for Regression Loss Meta-Learning

The proposed approach, `Loss2Net`, consists of a loss-function-agnostic regressor that performs a twofold learning: `Loss2Net` jointly learns (i) the output values that minimize a certain loss function (as any standard regressor) and (ii) *which* is the said loss function according to a posteriori system measurements. As discussed before, this concept finds applications in practical regression problems

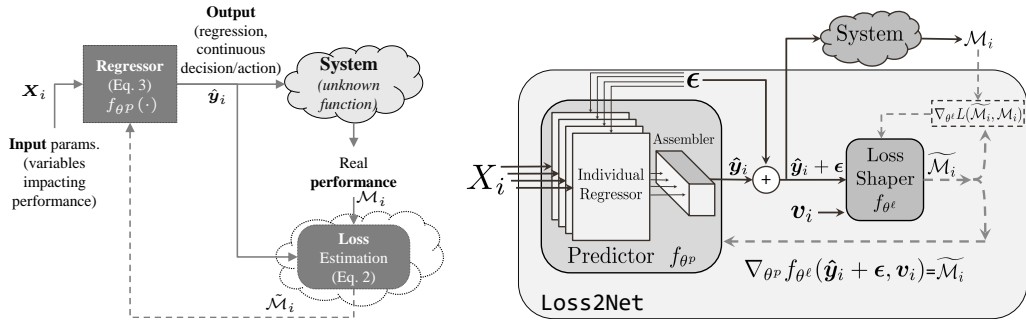

Figure 1: Problem overview.         Figure 2: Conceptual architecture of `Loss2Net` model.

where we can measure the performance resulting from the actions taken, but the expression of the objective cannot be characterized a priori. Next, we provide a formal definition of the problem; we then present the high-level concept of `Loss2Net` and its detailed architecture design and operation.

### 3.1 PROBLEM FORMULATION AND NOTATION

We denote the space of system state variables as $\mathcal{S} \subset \mathbb{R}^s$, the input space of the predictor as $\mathcal{X} \subset \mathbb{R}^{n_1 \times n_2}$, and the output space as $\mathcal{Y} \subset \mathbb{R}^m$. Thus, the predictor is modeled as $f_{\theta^p} : \mathcal{X} \to \mathcal{Y}$, where $\theta^p$ represents the predictor's parameters. For a given input matrix[2] $\boldsymbol{X}_i = (\boldsymbol{x}_{i,1}^\top, \ldots, \boldsymbol{x}_{i,n_2}^\top) \in \mathcal{X}$, the model makes a decision $\hat{\boldsymbol{y}}_i = f_{\theta^p}(\boldsymbol{X}_i) = (\hat{y}_{i,1}, ..., \hat{y}_{i,m}) \in \mathcal{Y}$, where $i$ represents the sample index.

We recall that the loss function represents the performance cost of the predictor's decision. Let us first consider the usual scenario in which the loss function is known, and let us denote this loss function as $f_{\mathcal{M}}$, such that the performance cost for sample $i$ is $\mathcal{M}_i = f_{\mathcal{M}}(\hat{\boldsymbol{y}}_i, \boldsymbol{v}_i)$, where $\boldsymbol{v}_i \in \mathcal{S}$ denotes the external observations that may impact $\mathcal{M}_i$. In such standard scenario, the predictor's objective is to minimize the performance cost of its decisions, i.e., to solve the following optimization:

$$\min_{\theta^p} \sum_i f_{\mathcal{M}}(\hat{\boldsymbol{y}}_i, \boldsymbol{v}_i) \quad \equiv \quad \min_{\theta^p} \sum_i f_{\mathcal{M}}\big(f_{\theta^p}(\boldsymbol{X}_i), \boldsymbol{v}_i\big). \tag{1}$$

However, we recall that the objective is characterized by an *a priori unknown* expression. Note that, even if this expression is not known, the performance is assumed to be measurable, and samples of $f_{\mathcal{M}}(\cdot)$ can be obtained by observing the outcome of the predictor's output on the system. We can define a second optimization problem that models the characterization of $f_{\mathcal{M}}(\cdot)$, which is uncharted at first. The solution of this optimization problem should casts an estimate of $f_{\mathcal{M}}(\cdot)$, denoted as $\tilde{\mathcal{M}}_i$, from the predictor decision $\hat{\boldsymbol{y}}_i$ and the current observations $\boldsymbol{v}_i$, i.e., $\tilde{\mathcal{M}}_i = f_{\theta^\ell}(\hat{\boldsymbol{y}}_i, \boldsymbol{v}_i)$, where $\theta^\ell$ represents the parameters to be optimized. Thus, the second optimization problem is defined as

$$\min_{\theta^\ell} \sum_i || f_{\theta^\ell}(\hat{\boldsymbol{y}}_i, \boldsymbol{v}_i), \; f_{\mathcal{M}}(\hat{\boldsymbol{y}}_i, \boldsymbol{v}_i) || \quad \equiv \quad \min_{\theta^\ell} \sum_i ||\tilde{\mathcal{M}}_i, \; \mathcal{M}_i||, \tag{2}$$

where $|| \cdot ||$ represents any distance-based metric, e.g., MAE or MSE, since the objective is now to mimic as closely as possible $f_{\mathcal{M}}(\cdot)$.

Consequently, when $f_{\mathcal{M}}(\cdot)$ is not known a priori, we cannot solve equation 1, and we have to optimize the predictor's decisions w.r.t. the estimated performance $f_{\theta^\ell}(\hat{\boldsymbol{y}}_i, \boldsymbol{v}_i)$, i.e., to solve

$$\min_{\theta^p} \sum_i f_{\theta^\ell}\big(f_{\theta^p}(\boldsymbol{X}_i), \boldsymbol{v}_i\big). \tag{3}$$

Fig. 1 shows the system operation and problem setting. While the literature usually deals with this scenario via iterative/alternate optimizations, `Loss2Net` *jointly* solves equation 2-equation 3.

### 3.2 LOSS2NET: MODEL CONCEPT AND TRAINING

The `Loss2Net` model aims at solving the problem above in an effective and general manner. To this end, `Loss2Net` contains two main blocks, as illustrated in Fig. 2: The first one, on the left,

---

[2]For the sake of clarity, we describe the predictor's input space as a regular $\mathbb{R}^{n_1 \times n_2}$ space. Nevertheless, for an input matrix $\boldsymbol{X}_i$ the proposed approach does not limit the different $\boldsymbol{x}_{i,k} \in \mathbb{R}^{n_1}$, $k \in \{1, \ldots, n_2\}$, to have the same size. We also use the terms *regressor* and *predictor* interchangeably throughout the document.

encloses the actual *predictor* that solves equation 3. The second block, on the right, represents the *loss shaper* block and handles equation 2. Thus, the *loss shaper* block receives as input the output of the first predictor block, and it is trained to return the objective performance that will be achieved by enacting such action, such that it acts as loss function for the *predictor* block. That is, this second block learns and encodes the initially unknown relationship between the performance metric and the decision of the model. Without loss of generality (w.l.o.g.), both blocks are built upon Deep Neural Networks (DNN). The whole process of jointly solving equation 3 is automated.

`Loss2Net` is an important aspect of the design of `Loss2Net` is the inner structure of the predictor. As depicted in Fig. 2, the predictor consists of a set of *individual regressors* (IRs) followed by one *assembler*. The rationale for this structure is that many practical management or engineering problems require the prediction of multiple system variables at once; such variables are often *intertwined*, i.e., take values that depend on each other. For instance, in multi-user resource allocation tasks, the system has a maximum physical capacity that cannot be exceeded, and predictions on the optimal resources reserved for each user are contingent on those booked for other users at the same time. Formally, in this exemplifying task, $X_i$ can be decomposed as a set of vectors $\{x_{i,j}\}_{j=1}^{n_2}$, one for each user, and the corresponding resource reservations $\hat{y}_{i,j}$ are inter-dependent. In this setting, the IRs are separate parallel layers that aim at predicting the future samples of each input variable; the assembler receives such estimated values and learns how they are both intertwined among them and related to the performance objective taught by the loss shaper. This structure improves the modularity of the solution, since each IR can be later used as pre-trained model for predicting a single variable $x_{i,j}$.

`Loss2Net` is a conceptual model, and its focus is not on improving the design of the predictor over state-of-the-art regression algorithms such as N-BEATS (Oreshkin et al., 2020), DeepAR (Salinas et al., 2020), PatchTST (Nie et al., 2023), N-HiTS (Challu et al., 2023), or TimesNet (Wu et al., 2023), but on addressing the loss-metric mismatch. In fact, the IRs blocks (cf. Fig. 2) can implement any prediction model, including those mentioned above. Yet, even a perfect predictor would not minimize a practical performance metric if trained with standard error-reduction losses.

As for the loss shaper, we adopt w.l.o.g. an Implicit Neural Representation (INR) architecture, which is motivated by the fact that this component does not aim at directly identifying the loss as a pure mathematical expression, but rather as a geometric multi-dimensional shape based on coordinates. It thus makes sense to implement it with a model intended for shape reconstruction. INR frameworks have shown significant capability (Sitzmann et al., 2020) to encode the functional relationship between a data sample and its coordinates (e.g., mapping a pixel position to its value) through a simple Multi-Layer Perceptron (MLP) architecture. INR frameworks were initially applied to 3D graphics and rendering (Sitzmann et al., 2019), and they have recently gained popularity for other domains, such as computer vision (Saragadam et al., 2023) or time series analysis (Dupont et al., 2022), thanks to their ability to perform compression, super-resolution, and handling of missing values.

Formally, an INR approximates the mapping from the coordinate space $\mathcal{T} \subset \mathbb{R}^n$ to the signal or feature space $\mathcal{X}(t) \subset \mathbb{R}^m$ (for a single coordinate $t \in \mathcal{T}$) via an MLP $f_\theta$ with weights $\theta$. As an example, in the image domain, the $\mathcal{T}$ space is the set of $(a, b)$ pixel locations, the $\mathcal{X}$ space is the RGB color space, and an image $X_i$ is the mapping $\mathcal{T} \to \mathcal{X}$. It can be extended to a multi-variate setting, where the coordinates $\mathcal{T}$ correspond to the different indexes, and the output $\mathcal{X}$ is the multi-variate feature space. Each element can be seen as a function $X_i$ from the coordinate space $\{j\}_{j=1}^J$ to the signal space and is given by a collection of pairs $d_i = \{(j, X_i(j))\}_{j=1}^J$ referring to both coordinates and values. Advanced INR models use the so-called SiREN layers (Sitzmann et al., 2020). SiREN layers contain a hyperparameter $\omega_0$ and are defined as $h^{l+1} = \sin(\omega_0(W^l h^l + b^l))$.

Some works on shape reconstruction prefer to use differentiable signed distance functions (SDFs) (Sitzmann et al., 2020) for shape reconstruction purposes. However, such approach is not suitable for the problem here considered, as it would imply either to know in advance the shape of the loss or to model it using purely random samples at first, which is not feasible for a real system. Therefore, in this work we consider instead the direct approach of learning the shape of the loss.

We make use of the Cyclic Learning Rate (CLR) method (Smith, 2017) to let the learning rate oscillate within a range whose extreme values are updated at each batch iteration. We found that this method largely reduces the sensitivity of `Loss2Net` to the initial configuration. Also, CLR accelerates the automatic convergence, which is especially useful in our context, where the loss function must be inferred during training.

---

**Algorithm 1:** Training procedure of `Loss2Net`

---

Initialize parameters $\theta^p$ of predictor NN $(f_{\theta^p}(\boldsymbol{X}_i, \boldsymbol{\epsilon}))$

Initialize parameters $\theta^\ell$ of loss-shaper NN $\left(f_{\theta^\ell}\left(f_{\theta^p}(\boldsymbol{X}_i, \boldsymbol{\epsilon}) + \boldsymbol{\epsilon}, \boldsymbol{v}_i\right)\right)$

Initialize predictor's learning rate $\alpha_t^p$, loss-shaper's learning rate $\alpha^\ell$, and noise std. deviation $\Sigma_t$

**for** $t = \{1, 2, ..., T_{training}\}$ **do**

    Randomly choose $\boldsymbol{\epsilon} = (\epsilon_1, ..., \epsilon_m)^\top$ from $\mathcal{N}_m(0, \Sigma_t)$

    Predict the output $\hat{\boldsymbol{y}}_{t+1} = f_{\theta^p}(\boldsymbol{X}_t, \boldsymbol{\epsilon})$

    Compute loss-shaper loss function: $\Lambda_{t+1} = L^2\left(f_{\theta^\ell}(\hat{\boldsymbol{y}}_{t+1} + \boldsymbol{\epsilon}, \boldsymbol{v}_{t+1}), f_{\mathcal{M}}(\hat{\boldsymbol{y}}_{t+1} + \boldsymbol{\epsilon}, \boldsymbol{v}_{t+1})\right)$

    Update loss-shaper weights: $\theta^\ell \leftarrow \theta^\ell - \alpha^\ell \nabla_{\theta^\ell} \Lambda_{t+1}$

    Update predictor weights: $\theta^p \leftarrow \theta^p - \alpha_t^p \nabla_{\theta^p} f_{\theta^\ell}(\hat{\boldsymbol{y}}_{t+1} + \boldsymbol{\epsilon}, \boldsymbol{v}_{t+1})$

    Update $\alpha_t^p$ and $\Sigma_t$

**end**

---

Concerning the `Loss2Net` concept and training, the following five important remarks are in order.

1. The meta-learning model outlined above is able to approximate a non-differentiable and non-continous objective $\mathcal{M}$ by a *differentiable and continuous alternative* $f_{\theta^\ell}(\cdot)$, which is implemented by the loss shaper INR upon training. This allows optimizing the predictor DNN under metrics that could not be directly used as losses, via a suitable approximation.

2. The input of the loss shaper INR is implemented by a different expression than that indicated in the formal problem definition of Section 3.1. Specifically, as portrayed in Fig. 2 and Algorithm 1, and detailed in Section 3.3, instead of providing the computed action $\hat{\boldsymbol{y}}_i$ to the loss shaper INR, we feed it with a disturbed version of $\hat{\boldsymbol{y}}_i$, by adding a random noise $\boldsymbol{\epsilon}$ that is also input to the predictor DNN. This methodological novelty allows *loss exploration* during training and helps the loss shaper to better model the target metric.

3. The `Loss2Net` model design allows *co-training* the predictor and loss shaper blocks during a same gradient descent iteration, so that each DNN is informed of (and can learn from) the improvements of the other. This fact makes the learned loss $f_{\theta^\ell}(\cdot)$ adapted to the inherent regression limits of the predictor, as later expounded in Section 3.4.

4. The structure in Fig. 2 allows for separated *pre-training* of each IR. Also, the IR-assembler architecture can be replaced by a single IR when not dealing with intertwined predictions.

5. `Loss2Net` naturally lends itself to support *transfer learning*. Indeed, the loss shaper block learns to characterize the unknown loss function that defines the relationship between the final performance and the metric space that outputs the predictor, and it does so irrespectively of the used regressor. Thus, the trained loss shaper block can be employed to train any predictor that faces the same complex problem, but it can be also used as an initial state for the loss shaper block in presence of different unknown functions that are expected to behave in a similar way. This property is assessed in Appendix D.3.

### 3.3 LOSS EXPLORATION

During training, the loss shaper block does not receive the exact value output by the predictor, $\hat{\boldsymbol{y}}_i$, but rather a disturbed version of it, $\hat{\boldsymbol{y}}_i + \boldsymbol{\epsilon}$, where $\boldsymbol{\epsilon} = (\epsilon_1, ..., \epsilon_m)^\top \sim \mathcal{N}_m(0, \Sigma_t)$, and $\Sigma_t$ evolves during the training process. The weight updates for both the predictor's DNN ($\theta_{t+1}^p$) and the loss shaper INR ($\theta_{t+1}^\ell$) are computed by gradient descent, as later explained in Section 3.4. `Loss2Net`'s algorithm is provided in Algorithm 1. The learning rates for predictor and loss shaper are, respectively, $\alpha_t^p$ and $\alpha^\ell$. The dependency of $\alpha_t^p$ on $t$ is due to the use of CLR mentioned in Section 3.2.

The noise $\boldsymbol{\epsilon}$ is only used in training, and it is set to 0 once the expression of the loss $f_{\theta^\ell}(\cdot)$ is learnt, during testing. In this regard, a critical design feature of `Loss2Net` is that $\boldsymbol{\epsilon}$ is also input to the regressor DNN: during training, it lets the prediction block learn the correlation between such input and the added disturbance to its output. Then, during inference, setting $\boldsymbol{\epsilon}$ to 0 allows producing outputs $\hat{\boldsymbol{y}}_i$ that are not biased by the loss exploration used in training.

The goal of the random variable $\epsilon$ is to allow for further exploration of the input values, supplying the loss shaper with a broader observation of the input domain beyond that provided by the training samples, and improving the reliability of the characterization of the loss function over the continuous domain. In fact, there is an intuitive analogy between this additive noise and the approach applied in reinforcement learning (RL) of taking a non-optimal typically random decision with some probability, creating an exploration-exploitation trade-off. Similar to what occurs in RL, we observe that the noise is most beneficial when it exponentially decays as the training advances. A detailed analysis of this aspect is presented in Appendix D.2 with data obtained from practical use cases.

### 3.4 CO-TRAINING OF PREDICTOR AND LOSS

`Loss2Net` is implemented as two cascaded DNNs, as illustrated in Fig. 2, which allows us to jointly optimize the two blocks through the same backpropagation process. The method is summarized in Algorithm 1, and the weights of the two DNNs are optimized during training as follows.

First, during the forward pass, the predictor $f_{\theta^p}$ is fed with a set of observations of the system state and outputs its decision $\boldsymbol{y}_i$. External observations $\boldsymbol{v}_i$ that may impact the metric, but not directly the prediction are measured and both are passed to the loss shaper, which computes the estimated performance function $\tilde{\mathcal{M}}_i = f_{\theta^\ell}\big(f_{\theta^p}(\boldsymbol{X}_i, \epsilon) + \epsilon, \boldsymbol{v}_i\big)$. At the same time, the actual performance of the decision $\mathcal{M}_i = f_{\mathcal{M}}\big(f_{\theta^p}(\boldsymbol{X}_i, \boldsymbol{\epsilon}) + \boldsymbol{\epsilon}, \boldsymbol{v}_i\big)$ is measured.

Then, the mismatch between estimated and true performance is evaluated via a legacy or standard loss function, and backpropagated first to the loss shaper INR. Here, the loss shaper updates its weights $\theta^\ell$ to better capture the relation between $\mathcal{M}_i$ and the combined values of the prediction $\hat{\boldsymbol{y}}_i$ and the system state $\boldsymbol{v}_i$. Within the same iteration, the updated loss is sequentially backpropagated to the predictor DNN, which allows improving the alignment of the prediction with the optimal decision that minimizes $\mathcal{M}_i$.

This design increases the efficiency of the training phase with respect to the case where each block is optimized independently, e.g., by feeding the loss shaper block with random predictions and, once the loss has been learned, using it to train the predictor. Indeed, co-training allows learning a loss $f_{\theta^\ell}(\cdot)$ that is adapted to the intrinsically limited accuracy of the predictor; as an example, co-training may lead to learning diverse shapes of the loss depending on the magnitude of the target variable if the quality of the prediction is found to be affected by the absolute value of the target variable. We will observe practical situations where this type of adaptation occurs in Section 4.

It is worth noting that such a co-training represents a major novelty of our model with respect to previous related proposals (Sung et al., 2017; Wu et al., 2018; Grabocka et al., 2019). Indeed, the end-to-end backpropagation training was not possible in prior models, and the two elements (the learning-to-act and the learning-to-correct blocks) were trained iteratively or in a nested manner.

## 4 EVALUATION AND APPLICATION USE CASES

We perform extensive analyses to evaluate the performance and properties of `Loss2Net`. For that, we study different aspects and advantages of the method; in particular, we evaluate its hyper-parameter sensitivity, its transfer learning potential, the intrinsic explainability that offers `Loss2Net`, and we perform an ablation study. The details of these analyses are provided in Appendices C-D, while we only briefly present here the main takeaways due to space limits.

**Sensitivity to hyper-parameters.** Both the adaptive learning rate and the loss exploration noise make the model robust to hyper-parameters, as later explained in Appendix D. In addition, the flexibility of the `Loss2Net`'s design allows us to select robust architectures because it can accommodate any internal architecture of the predictor and the loss shaper.

**Ablation study.** We show how all of the components of our design provide important contributions to achieving high predictive performance by demonstrating a performance reduction when systematically removing blocks or modifying the model design, e.g., as proposed by Grabocka et al. (2019).

**Transfer learning.** We demonstrate the potential of our design for transfer learning, and thus for fine-tuning or few-shot multi-task learning, using different settings for one additional use case: several network data centers serving different user populations and experiencing diverse demands.

**Explainability** In contrast to other techniques that are expected to be able to address this kind of problems, such as RL techniques, we can obtain explainability of those results. In fact, we can easily represent the loss that the loss shaper models in order to understand the results of the predictors.

Next, we provide a detailed evaluation of `Loss2Net` for two practical use cases in system engineering where the target performance is an unknown function of the predictor's output: (i) smart power grid management, and (ii) a full-fledged use case which targets a real-world system management scenario where intertwined predictions are required to execute admission control and resource allocation on concurrent user demands. We employ measurements from real-world production systems for the user demands, and imitate entangled relations between the prediction and the performance metric via suitable emulation environments.

### 4.1 Experimental Datasets

The first dataset describes the hourly energy consumption in part of the Eastern Interconnection grid in the United States of America between 2002 and 2018. The data comes from a regional transmission organization (RTO) (Mulla, 2018) and incorporates information from several power providers, each managing a distinct geographic area. The dataset contains over 145,000 samples.

The second dataset describes the mobile data traffic demand generated by 12 applications, which include video streaming, social networking, and web browsing services. The data represents the traffic demand from several millions of subscribers in a large metropolitan area during several consecutive months (Marquez et al., 2017). The data was collected by the mobile network operator, resulting in traffic loads (in bytes) every five minutes, for a total of more than 22,000 samples per application.

### 4.2 Use Case I: Power Grid Management

**Metric Definition:** The complex field of power grid management and the study of smart grids are ruled by a significant number of diverse Key Performance Indicators (KPI) (IEEE, 2012; Personal et al., 2014). One of the fundamental dimensions that define the performance in such scenarios is the reliability of the network, i.e., how often the network fails to provide the required power. Interestingly, the reliability in power management is not only measured by the *frequency* of power cuts due to the under-provisioning, but also by the *duration* of these cuts (IEEE, 2012). The smart grid manager is especially interested in preventing long-term under-estimations; that is, the performance metric incorporates memory about past outages. Specifically, in the case of overdimensioning, the cost scales linearly with the excess estimated power. For underestimations, if the previous prediction was also underestimated, the current cost is added to the precedent cost in a recursive manner. Thus, the metric depends on the previous system state, and involves discrete (thus discontinuous) jumps when consecutive outages occur. Manually designing a surrogate version of the multi-dimensional loss function matching the metric above is not trivial, and we are not aware of any such attempt in the literature. `Loss2Net` removes this barrier by learning the correct loss in an automated way.

**Network architecture:** In this experiment the predictor $f_{\theta^P}$ used is a multi-layer RNN regressor. The INR $f_{\theta^\ell}$ is defined using three inputs: $\boldsymbol{y}_{t+1}$, $\hat{\boldsymbol{y}}_{t+1} = f_{\theta^P}(X_t)$, and the number of previous successive under-provisioned samples $\boldsymbol{v}_t$ (maxed at 5), which is a time-dependent variable. Details on the network architecture are provided in Appendix B.

**Comparison:** Recent studies in the computer science literature have proposed an expert-designed loss function, termed $\alpha$-OMC (Bega et al., 2019), which can be considered a handcrafted, differentiable surrogate of the true performance metric. We employ $\alpha$-OMC as a state-of-the-art loss benchmark. Although initially crafted for network-related applications, its underlying principle remains consistent in our context: tolerating a minimal level of surplus to minimize disruptions. Yet, $\alpha$-OMC cannot incorporate the temporal aspect of the cost performance metric.

**Results:** We summarize the evaluation results in Table 1, where we show the average and standard deviation of the operator's normalized cost (i.e., the metric). We observe that `Loss2Net` significantly outperforms traditional training methods based on standard loss functions, eliminating 96% of the cost. While the advantage over generic loss roots in the loss-metric mismatch and is easily understood, the improvement with respect to $\alpha$-OMC is explained as `Loss2Net` effectively adapts to varying demands. The model learns a loss that compensates for different accuracies in predicting power consumption values of diverse magnitudes. Further details are presented in the Appendix.

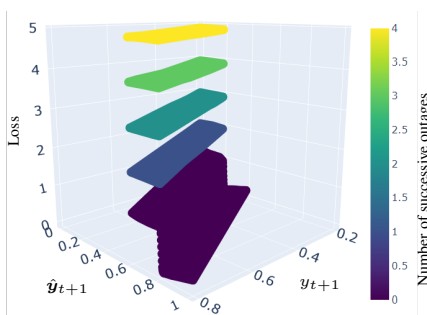

Figure 3: 4-D shape of the *learned* loss.

Table 1: Normalized performance results for the power grid scenario (use case I) when the same predictor DNN is trained with 4 different loss functions: MSE, MSLE, the proposed `Loss2Net` model, and $\alpha$-OMC, an expert-designed surrogate loss from Bega et al. (2019).

| Loss function | Normalized cost (metric) | $\frac{\text{Cost}}{\text{Loss2Net Cost}}$ |
|---|---|---|
| MSE | $1.804\pm0.262$ | $\times32.21$ |
| MSLE | $1.813\pm0.253$ | $\times32.37$ |
| Loss2Net | $\mathbf{0.056\pm0.002}$ | $\times1$ |
| $\alpha$-OMC | $0.114\pm0.007$ | $\times2.04$ |

In this first use case, we are particularly interested in presenting how `Loss2Net` can adapt to a recursive cost relationship. Fig. 3 shows the learned loss function: the dependency on the extra dimension representing the number of consecutive outages is correctly captured, as shown by the color scale in the plot. The result shows the potential of the proposed approach to characterize unknown and non-trivial loss functions for regression problems. In this particular case, `Loss2Net` learns a differentiable approximation of a step-wise discrete function.

## 4.3 USE CASE II: JOINT ADMISSION CONTROL AND RESOURCE ALLOCATION

We consider the task of admission control and resource allocation (AC-RA), which is a problem that appears in many different fields and business cases. Consider an operator controlling a certain amount of resources and a set of clients that want to get access to some of those assets paying for that access. The operator must decide to whom it offers such resources and the amount of resources dedicated to each client. Consequently, the problem is defined by (i) the total amount of resources to be shared, (ii) the amount of resources that each client requests, (iii) the price of a resource unit, and (iv) the operator's cost from operating the system. Now, we consider the second dataset described in Section 4.1 for network traffic loads, since AC-RA is routinely required in communication networks, where the resources are the data transport assets. This scenario considers a more complex task that deals with different predictors trying to compete to access the same resources.

The perspective of the analysis is that of maximizing the network operator's profit. The operator allocates customized resources *in advance* based on expected traffic loads from each client. The operator's revenue depends on the QoE that the users of each client experience. AC-RA problems also open the door to yield management techniques such as overbooking, since in many situations the clients request a certain amount of resources that later are not necessarily utilized.

**Metric Definition:** We consider that the deal between client and operator is defined by a contract that sets the following rules. The contract determines the target performance, measured in terms of the requested end-user QoE, specifies the price that the client must pay, which is often proportional to the requested traffic load, and defines the fees and price reductions stemming from a violation of the terms of conditions. For example, the final price is proportionally reduced if the QoE drops below a minimum acceptable threshold. Furthermore, the operator incurs operating expenses that are proportional to the quantity of reserved resources.

Importantly, the operator is not capable of knowing what is going to be the end-user's QoE. First, because it cannot measure it (as it is an information that is acquired by the service provider, i.e., the client); second, because the relationship between QoE and network resources is known to be an obscure function that cannot be easily modeled. Besides this, even in the hypothetical case in which such function was known, the AC-RA problem is an NP-hard problem, with the non-trivial trade-off between over-allocating resources (such that the QoE is ensured at the expense of increased operating costs) and under-allocating them (such that the operator can *overbook* the system, accepting more clients and hence earning more money, at the risk of not reaching the QoE levels, which would lead to service outages and a huge cost in terms of customer churn).

**Solution based on `Loss2Net`:** In this scenario, where the network operator cannot analytically know the relation between the resource reservation and the end-users' QoE, we can apply `Loss2Net`

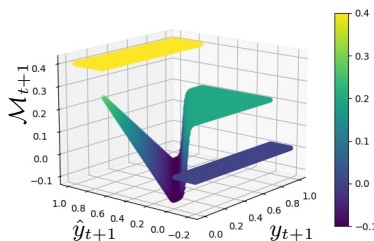

Figure 4: *Learned* performance cost sample when the network serves *a single* client.

Table 2: Evaluation for use case II. We present the normalized operator's profit (i.e., the performance metric) and the mean amount of clients accepted.

| Clients | Loss2Net | | knapsack | |
|---|---|---|---|---|
| | Profit | Accepted | Profit | Accepted |
| 2 | **0.1195** | **1.28** | 0.0430 | 1.00 |
| 4 | **0.2382** | **2.76** | 0.1620 | 2.60 |
| 6 | **0.3827** | **5.67** | 0.2661 | 4.36 |
| 8 | **0.4215** | **7.10** | 0.3496 | 4.83 |
| 12 | **0.5424** | **10.80** | 0.3374 | 8.94 |

in order to obtain both (i) which clients are accepted at any decision time slot, and (ii) the amount of resources reserved to each of the accepted clients, while simultaneously capturing the previously unknown relationship between resource allocation and QoE performance.

In particular, the input to Loss2Net consists of time series of the past traffic measurements for each one of the clients, and the output is the allocated resources in terms of capacity reservation for each one of the clients (with zero resources indicating that the client has not been accepted). To illustrate the complexity of the meta-learning task, we provide in Fig. 4 the expression of the loss function in the naive case of a single client, which is the only case that can be depicted in only three dimensions. Yet, we can observe how even in such toy example the shape of the relationship is tangled and also highly sensitive to the input value, and hence the corresponding practical multidimensional scenario requires strong meta-learning capabilities. More details are presented in the Appendix.

**Benchmark:** Since the performance metric is unknown a priori, we cannot compare with approaches that are based on standard loss functions. Because of that, we proceed to show the results of an ablation study in which we show the importance of the inner structure of the predictor in Fig. 2, by comparing Loss2Net with a solution where the forecast is done on each individual client. The predictors' output for the RA of each client is then used as input data to solve an optimization program. Formally, this optimization problem is given by

$$\max_{x_s(t)} \sum_{s \in \mathcal{S}} R_s(t) x_s(t) \quad \text{s.t.} \sum_{s \in \mathcal{S}} c'_s(t) x_s(t) \leq C. \quad (4)$$

where $x_s \in \{0, 1\}$ is the binary variable for admission control, $\mathcal{S}$ the set of clients, $R_s$ is the revenue obtained from client $s$, $C$ is the total amount of resources and $c'_s$ is the predicted resource allocation output by the predictor block of client $s$. This is in fact the well-known NP-hard Knapsack problem (KP), and we refer to its solution as knapsack. We show that the latter architecture do not perform well when there are intertwined prediction scenarios. Note that the intertwining comes from the fact that the predicted resources do not depend only on the past samples of the client's traffic load, but also on the decision of resource allocation of the other clients and the total amount of resources.

**Results:** We provide a quantitative evaluation in Table 2. It shows the final performance values for Loss2Net and knapsack for different sizes of the clients' set $\mathcal{S}$. In summary, Loss2Net is able to provide an increase of profit for the operator up to 60% when all 12 clients are present, accepting in average 2 extra clients with respect to knapsack. It is important to remark that such improvement is achieved with no prior knowledge of the system nor of the objective, which shows its meta-learning capabilities and its practical advantage over current approaches.

## 5 CONCLUSIONS

We propose Loss2Net, a meta-learning model for unknown regression losses, and unveil how this previously overlooked approach is in fact effective in practical scenarios where the relationship between model predictions and the target performance is not known a priori. Experiments with real-world heterogeneous datasets prove how Loss2Net can successfully learn losses that capture complex, time-correlated and non-differentiable metrics. Our analysis also highlights how co-training prediction and loss allows "learning more than the teacher knows", since the loss is tailored to the inherent and specific (in)accuracy of the predictor. This property allows us to automatically adapt the loss to the performance of the predictor and provides us with high-dimension loss functions that better capture the corresponding problem with respect to standard uni-dimensional losses.

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

## A APPENDIX STRUCTURE

Next, we present complementary results, which aim at providing a more in-depth vision of the `Loss2Net` approach and its potential. In particular, we provide a more detailed description of the approach and deeper analysis including an ablation study and an additional use case that helps us show that capabilities on transfer learning and explainability of the approach. First, we present the detailed architecture of the neural networks considered in Appendix B. Then, Appendix C presents the ablation study on our proposed approach.

## B NEURAL NETWORK ARCHITECTURES

We detail in this appendix the architecture of the different neural networks considered, for the sake of reproducibility. For both use cases, we employ a simple triangular version of CLR, with 3 cycles across the full training phase.

In use case I, the predictor is composed of 3 layers using respectively (128, 128, 64) LSTMs cells and ReLU as activation function, and one output layer. The loss shaper has 3 dense layers, with respectively (256, 256, 128) units in it using the SiREN activation function where we set $\omega_0 = 30$ for the first layer and $\omega_0 = 1$ for all other layers.

In use case II, the predictor is composed of up to 12 IRs depending of the number of services with each composed of 3 layers using respectively (128, 128, 64) LSTMs cells with ReLU as activation function. The assembler is composed of 4 dense layers of respectively (256, 256, 128, 64) neurons with ReLU as activation function. The loss shaper has 5 dense layers, with respectively (256, 256, 256, 256, 128) units in it using the SiREN activation function where we set $\omega_0 = 30$ for the first layer and $\omega_0 = 1$ for all other layers. This loss shaper has a more complex architecture than in use case I as the metric to model is more complex than the one in use case I.

For the additional use case III depicted below in Appendix D, the predictor is composed of 3 layers using respectively (128, 128, 64) LSTMs cells and ReLU as activation function, and one output layer. The loss shaper has 3 dense layers, with respectively (256, 256, 128) units in it using the SiREN activation function where we set $\omega_0 = 30$ for the first layer and $\omega_0 = 1$ for all other layers.

## C ABLATION STUDY

Next, we proceed with an ablation study. To this end, we consider a toy regression example: the unknown objective is that of producing an average of all inputs in the next time slot. Formally, we consider $n_{in}$ different time series, and the predictor shall output a scalar $\hat{y}_{t+1} = \frac{1}{n_{in}} \sum_{i=1}^{n_{in}} x_{t+1}^{(i)}$. This is a simple problem with intertwined predictions, as the output depends on forecasts of all inputs. We consider uncorrelated time series, to prevent degenerate cases where the average could be artificially simplified as $y_{t+1} = K\boldsymbol{X}_t$. We present, in the following, the list of different models that we consider for the ablation study.

- `ReLu` identifies the impact of considering the ReLu activation function instead of the SiREN activation function for the loss shaper.

- The `bm-monolithic` approach follows the `Loss2Net` structure but without the inner structure of the predictor illustrated in Fig. 2 of the main text. `Loss2Net`'s structure splits the predictor in a first block of parallel IRs that connects to an assembler. Instead, `bm-monolithic` uses a single block with LSTM and MLP layers to implement the predictor, which gets the whole input $\boldsymbol{X}_t$ and outputs the forecast $\hat{\boldsymbol{y}}_{t+1}$.

- The `bm-split` design decomposes the original multi-dimensional problem along single input variables: it runs $n_{in} = n_{out}$ different and independent predictors, i.e., indepen-

Table 3: Results for the controlled experiment where the goal is learning that the loss function is the MAE of the estimate w.r.t. $y_{t+1} = \frac{1}{n_{in}} \sum_{i=1}^{n_{in}} x_{t+1}^{(i)}$. Values in the table represent the average MAE $\pm$ its std. deviation, for various number of input variables $n_{in}$. We evaluate our model and the benchmarks described in Appendix C.

| $n_{in}$ | Loss2Net | ReLu | bm-merged | bm-split | bm-monolithic |
|---|---|---|---|---|---|
| 2 | **0.018±0.014** | 0.024±0.012 | 0.101±0.081 | 0.119±0.094 | 0.067±0.042 |
| 4 | **0.020±0.012** | 0.023±0.010 | 0.095±0.073 | 0.116±0.089 | 0.047±0.026 |
| 6 | **0.018±0.010** | 0.021±0.008 | 0.098±0.076 | 0.090±0.073 | 0.043±0.026 |
| 12 | **0.010±0.006** | 0.012±0.003 | 0.093±0.079 | 0.088±0.096 | 0.038±0.024 |
| $n_{in}$ | noNoise | fixedNoise | fixedLR | expLR | Grabocka |
| 2 | 0.025±0.010 | 0.024±0.019 | 0.028±0.013 | 0.024±0.013 | 0.029±0.015 |
| 4 | 0.026±0.010 | 0.025±0.017 | 0.026±0.013 | 0.022±0.011 | 0.026±0.012 |
| 6 | 0.023±0.009 | 0.022±0.012 | 0.024±0.009 | 0.022±0.008 | 0.022±0.011 |
| 12 | 0.015±0.003 | 0.013±0.006 | 0.013±0.004 | 0.013±0.004 | 0.015±0.006 |

      dent instances of `Loss2Net`, each one attempting to learn the loss-function governing its corresponding variable unaware of the other predictors.

- The `bm-merged` architecture lies between the previous two benchmarks. It employs multiple parallel predictors that feed a single loss shaper during training.

- The `noNoise` design, which consists in setting $\epsilon = 0$ in `Loss2Net`'s training.

- The `fixedNoise` design, in which the variance of the noise $\epsilon$ is fixed throughout the training. This variance is set to be $3\%$ of the average value of the data, and this specific number is picked as giving the best results with a fixed amount of noise.

- We also check the impact of CLR, by evaluating `fixedLR`, where the learning rate is fixed, and `expLR`, where we apply exponential decay learning rate.

- Finally, we consider `Grabocka`, an approach inspired from Grabocka et al. (2019). `Grabocka` uses a similar training approach as the bilevel programming optimization of the model parameters adopted by Grabocka et al. (2019), where the loss model and predictor are alternatively trained for some epochs, instead of being trained simultaneously in the same gradient descent iteration as in `Loss2Net`.

Table 3 reports the corresponding results, where we measure the MAE with respect to the true value, i.e., the actual average. `Loss2Net` outperforms all the other models for all possible scenarios. We observe how the additive noise with exponential decay and the CLR help `Loss2Net` to improve the performance, which is also more uniform throughout all the cases, although the performance of `noNoise`, `fixedNoise`, and `expLR` is not far away. `fixedLR` lies behind, probably due to its dependence on the initial value of the LR. Note that the additive noise with exponentially decaying variance is not only helpful to increase the performance, but also to speed up the learning curve.

With respect to `bm-split` and `bm-merged`, they clearly underperform with respect to `Loss2Net`, which is able to reduce the error generated by these approaches between 75% and 90%. This shows that decoupling each one of the variables considerably impacts the performance, and proves that the inner structure of the predictor, with the separated IRs that are fed into the assembler, is crucial to combine the contribution of each variable and ensure that the task is learnt. The `bm-monolithic` model, which does not consider individual IRs, also underperforms with respect to `Loss2Net`.

Finally, the simultaneous training of loss shaper and predictor through the same backpropagation iteration improves the performance up to 35%, as observed by comparing `Loss2Net` vs. `Grabocka`.

## D    ADDITIONAL USE CASE AND MODEL ANALYSES

The objective of this section is to discuss additional aspects of the proposed model, namely ($i$) parameter sensitivity analysis, ($ii$) potential for transfer learning, and ($iii$) explainability. For this, we introduce a new use case that represents a simpler version of use case II from Section 4.3.

Table 4: Performance results for three different services (Facebook, Twitch, and Youtube), for four different datacenters and four different loss functions.

| Datacenter D1 (values $\times 10^{-1}$) | | | |
|---|---|---|---|
| Loss function | Facebook | Twitch | YoutTube |
| MSE | 4.651±0.748 | 4.741±0.502 | 4.867±0.685 |
| MSLE | 4.503±0.697 | 4.405±0.3873 | 4.478±0.369 |
| Loss2Net | **0.877±0.013** | **1.354±0.014** | **1.188±0.003** |
| Surrogate $\alpha$-OMC | 1.186±0.015 | 1.685± 0.018 | 1.563±0.023 |

| Datacenter D2 (values $\times 10^{-1}$) | | | |
|---|---|---|---|
| Loss function | Facebook | Twitch | YoutTube |
| MSE | 4.095±0.790 | 4.265±0.625 | 4.709±0.319 |
| MSLE | 4.203± 0.906 | 4.385± 0.348 | 5.060±0.628 |
| Loss2Net | **0.880±0.029** | **1.361±0.004** | **1.302±0.008** |
| Surrogate $\alpha$-OMC | 1.151±0.010 | 1.632± 0.016 | 1.497± 0.008 |

| Datacenter D3 (values $\times 10^{-1}$) | | | |
|---|---|---|---|
| Loss function | Facebook | Twitch | YoutTube |
| MSE | 4.567±0.637 | 4.175±0.286 | 4.591±0.226 |
| MSLE | 4.469±0.961 | 4.119±0.304 | 5.185±0.607 |
| Loss2Net | **1.102±0.031** | **1.376±0.010** | **1.480±0.016** |
| Surrogate $\alpha$-OMC | 1.365±0.0192 | 1.555±0.023 | 1.683±0.018 |

| Datacenter D4 (values $\times 10^{-1}$) | | | |
|---|---|---|---|
| Loss function | Facebook | Twitch | YoutTube |
| MSE | 4.751±0.631 | 4.492±0.523 | 4.620±0.540 |
| MSLE | 4.513 ±0.598 | 4.434±0.603 | 4.738±0.625 |
| Loss2Net | **0.940±0.013** | **1.601±0.010** | **1.305±0.012** |
| Surrogate $\alpha$-OMC | 1.230±0.014 | 1.712± 0.016 | 1.483±0.017 |

### D.1 USE CASE III: PREEMPTIVE NETWORK RESOURCE ALLOCATION

We make use of the second dataset presented in Section 4.1, which was also considered in Section 4.3. We recall that this dataset contains the mobile data traffic demand generated in a large metropolitan area during several consecutive months, and it includes data from different network *datacenters*, each serving the traffic transiting at a different set of communication antennas. We consider four mobile services: Netflix, Facebook, Twitch and YouTube; in addition, we consider heterogeneous datacenters, each serving a geographically separated group of communication antennas, what will be later explained more in detail in Appendix D.3. Each combination of service and datacenter entails dissimilar patterns in the user demand for network resources Marquez et al. (2017).

**Metric:** The goal of the operator at time $t$ is forecasting the required network resources (e.g., data transport capacity) needed to serve the demand for the service in the following 5-minute time step — an interval compatible with the operation time scale of modern orchestrators of network functions Gil Herrera & Botero (2016). In this case, the performance metric is asymmetric: the operator seeks to ($i$) avoid an expensive monetary fee $\beta$ owed to the service provider in case insufficient resources are allocated and the future traffic $\boldsymbol{y}_{t+1}$ cannot be served, and ($ii$) prevent unnecessary overdimensioning beyond $\boldsymbol{y}_{t+1}$ in case of prediction errors with a positive sign. The performance metric $\mathcal{M}_{t+1}$ is thus recorded as the actual economic cost incurred by the operator due to the forecast $\hat{\boldsymbol{y}}_{t+1}$. We note that the task of resource allocation under an asymmetric performance metric is common across a wide range of modern systems dealing with the planning and management of a limited amount of resources, including, e.g., retailing, transport, or logistics.

**Network architecture:** In this experiment, we employ as the predictor $f_{\theta^P}$ a multi-layer RNN regressor. The loss shaper neural network $f_{\theta^\ell}$ is an MLP with the simplest possible case for Loss2Net, using only $\boldsymbol{y}_{t+1}$ and $\hat{\boldsymbol{y}}_{t+1}$ as input. The predictor RNN is also trained using the $\alpha$-OMC loss function for comparison purposes. A detailed description of the neural network is presented in Appendix B.

**Results:** We provide in Table 4 the performance results for the preemptive network resource allocation problem described above, summarizing the performance figures of all services in separate tables for each datacenter. Results are aligned with the ones presented in Section 4, with an important gap between standard MSE and MSLE losses and more advanced techniques. In all cases, Loss2Net outperforms the expert-designed surrogate loss $\alpha$-OMC by 10% in the worst scenario, *even without prior knowledge of the relationship between prediction and performance metric*.

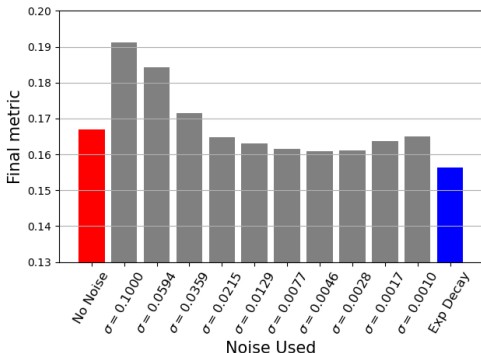 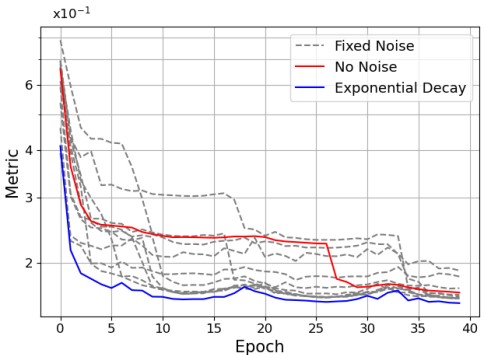

Figure 5: Loss metric results as function of the exploration noise's variance. For the exp. decay case, variance decreases as the epochs advance.

Figure 6: Learning curve for different exploration noise's variances. Grey lines correspond to the fixed variances shown in Fig. 5.

### D.2 SENSITIVITY TO HYPER-PARAMETERS

The model's hyper-parameters include the neural network architectures implementing the general concept in Fig. 2, the learning rate, and the loss exploration noise $\epsilon$. Next, we discuss the robustness to each such hyper-parameter.

**Neural network architecture.** The design of the neural networks that implement the predictor (IRs and assembler) and loss shaper blocks is specific to the target use case. This is an intrinsic property of any data-driven model, and is orthogonal to the general Loss2Net concept. We report the exact settings of the neural networks used in our experiments in Appendix B. The ablation study in Appendix C shows that, while the internal structure of the three main blocks of Loss2Net can be adapted to each particular case, the main structure composed by these blocks (IRs, assembler, and loss shaper) is a basic requirement and needs to be maintained.

**Learning rate.** We employ CLR, which is known to reduce the sensitivity of models to the choice of learning rate. The ablation study presented earlier in Appendix C shows how this is beneficial in most settings.

**Loss exploration noise.** We make the noise decay exponentially over time. This ($i$) avoids the need to fine-tune a fixed noise hyper-parameter, and ($ii$) improves the overall performance with respect to any fixed noise value. To demonstrate that this is the case, we consider a zero-mean Gaussian noise, and analyze the performance for different values of the noise variance. First, we consider the case with no noise (i.e., zero variance), which serves as a benchmark and shows the performance of a baseline model without this parameter. Later, we consider ten different variances, such that the standard deviation follows a geometric progression from 0.001 to 0.1. This extreme values are selected as meaningful range after a detailed evaluation. Finally, we consider the exponential decay case in which the variance decreases as the number of epochs increases, borrowing the idea from the exploration-exploitation trade-off of RL Sutton & Barto (2018). In particular, we consider that the standard deviation at epoch $t$ is equal to $\frac{0.2}{(0.1t)^\phi}$, where $\phi$ is a parameter selected to ensure that the value at the first epoch is 0.1 and that at the last epoch is 0.001, for a fair comparison with the fixed-variance cases.

Results are shown in Fig 5 and Fig. 6. In Fig. 5, we represent the final metric evaluated for the different noise variance cases, averaged over several realizations. We can observe how the results are quite sensitive to the variance of the noise, and adding noise can also be detrimental, whereas the exponential decay case obtains the best performance. This last adaptive approach is beneficial not only because it achieves the best performance, but also because it allows us to avoid the hyperparameter tuning required in view of the sensitiveness of the results to the variance. In Fig. 6, we show the learning curve (also averaged over the different training realizations), where we only highlight the no-noise and the exponential decay curves for the ease of visualization. The exponential decay approach provides a much faster and stable convergence. It has also been observed that it reduces the probability of obtaining a diverging solution.

Table 5: Transfer learning results, when run in different network datacenters serving diverse Facebook traffic demands. All values are $\times 10^{-1}$.

| Predictor trained at↓ | Metric est. trained at D1 | Metric est. trained at D2 | Metric est. trained at D3 | Metric est. trained at D4 |
|---|---|---|---|---|
| D1 | **0.877±0.013** | 1.028±0.050 | 1.021±0.021 | 1.044±0.040 |
| D2 | 1.002±0.048 | **0.880±0.029** | 1.001±0.036 | 1.035±0.047 |
| D3 | 1.226±0.012 | 1.243±0.050 | **1.102±0.031** | 1.245±0.023 |
| D4 | 1.088±0.037 | 1.081±0.028 | 1.076±0.022 | **0.940±0.013** |

Table 6: Transfer learning results, when run in different network datacenters serving diverse Twitch traffic demands. All values are $\times 10^{-1}$.

| Predictor trained at↓ | Metric est. trained at D1 | Metric est. trained at D2 | Metric est. trained at D3 | Metric est. trained at D4 |
|---|---|---|---|---|
| D1 | **1.354±0.014** | 1.522±0.012 | 1.497±0.012 | 1.530±0.016 |
| D2 | 1.479±0.005 | **1.361±0.004** | 1.489±0.008 | 1.490±0.015 |
| D3 | 1.487±0.014 | 1.487±0.017 | **1.376±0.010** | 1.491±0.024 |
| D4 | 1.671±0.014 | 1.670±0.015 | 1.684±0.011 | **1.601±0.010** |

Table 7: Transfer learning results, when run in different network datacenters serving diverse Youtube traffic demands. All values are $\times 10^{-1}$.

| Predictor trained at↓ | Metric est. trained at D1 | Metric est. trained at D2 | Metric est. trained at D3 | Metric est. trained at D4 |
|---|---|---|---|---|
| D1 | **1.188±0.003** | 1.401±0.013 | 1.382±0.018 | 1.389±0.021 |
| D2 | 1.433±0.006 | **1.302±0.008** | 1.423±0.013 | 1.443±0.017 |
| D3 | 1.693±0.016 | 1.684±0.021 | **1.480±0.016** | 1.686±0.022 |
| D4 | 1.396±0.013 | 1.345±0.022 | 1.388±0.020 | **1.305±0.012** |

## D.3 TRANSFER LEARNING

As the traffic serviced by each base station can differ substantially, each datacenter experiences quite diverse temporal dynamics of the demand Marquez et al. (2017). Yet, the performance metric is the same for all datacenters, and the resource allocation predictors dedicated to each datacenter shall all be trained to optimize it.

This is an ideal setting where to test the capability of `Loss2Net` for transfer learning. Specifically, we evaluate how a predictor trained with the `Loss2Net` approach (i.e., by jointly training the predictor and the loss shaper block) performs with respect to a predictor that is trained with a not-varying loss shaper block that has been previously trained with the data from another datacenter. It is important to note that the loss shaper block could also be trained with an initialization that uses an already trained configuration (which is usually known as fine-tuning or few-shot meta-learning), in order to adapt to the predictor behavior; this case is depicted in Appendix E.

Results are shown in Tables 5, 6 and 7. Each column indicates the datacenter in which the loss shaper block has been trained, whereas the rows indicate the datacenter for which that same loss shaper block, once trained for the column datacenter data, is used to train from scratch the predictor. Quite naturally, the values on the diagonal reflect the best performance, as they refer to cases where the full `Loss2Net` model trained on the traffic measured at a specific datacenter is employed to predict new demands in the same datacenter. However, by looking at values beyond the diagonal, where a loss function trained on traffic in one datacenter is used to train a new predictor for a different datacenter, performance remains on the same level. Specifically, by comparing the values in Table 5 with those in Table 4, transfer learning via `Loss2Net` yields performances in new datacenters that are still better than or comparable with those obtained with the expert-designed $\alpha$-OMC function in use case I. Also, the same performance remains much better than that of legacy mismatched losses like MSE or MSLE.

## E EXPLAINABILITY

In this section, we explore how the results from `Loss2Net` are interpretable, setting it apart from other methods like Reinforcement Learning. A significant strength of `Loss2Net` is its ability to

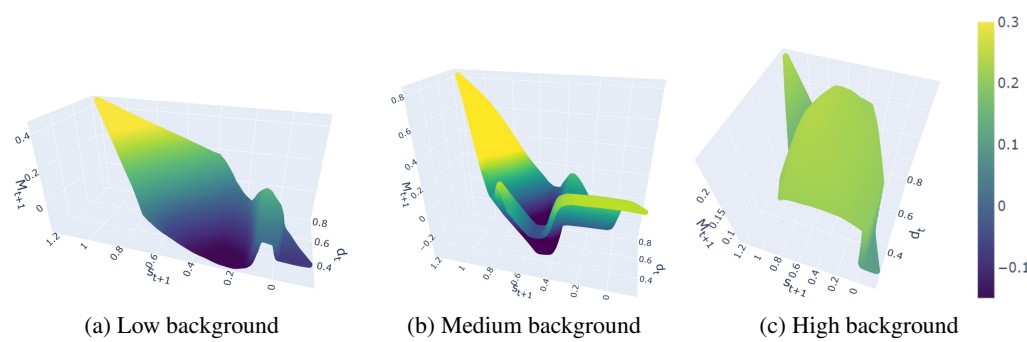

(a) Low background  (b) Medium background  (c) High background

Figure 7: Learned loss for the AC-RA task in sliced networks. We show the learned cost (z axis) for varying anticipatory decisions (x axis) and traffic demands (y axis) of one slice, while all other slices have fixed values. We represent the cases where the other slices generate (a) low, (b) medium, and (c) high traffic loads.

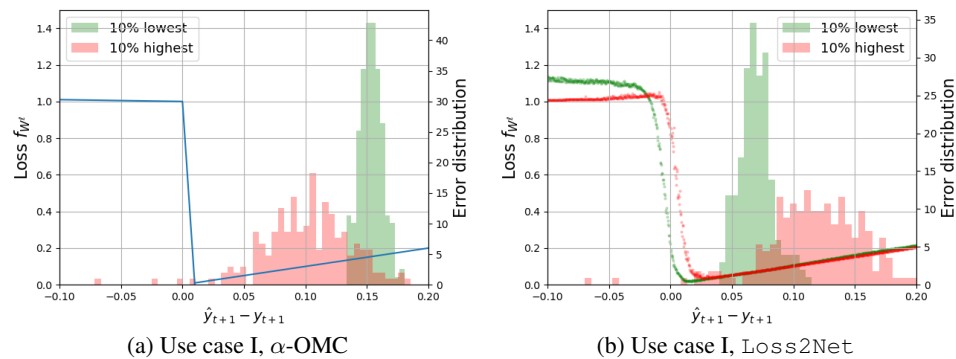

(a) Use case I, $\alpha$-OMC  (b) Use case I, Loss2Net

Figure 8: Loss function and error distributions for the top and bottom 10% of predicted values, for the network resource allocation to the traffic service under (a) expert-designed $\alpha$-OMC loss and (b) the proposed Loss2Net.

easily visualize the learned loss by the loss shaper. This is exemplified in Fig. 3 for use case I. The model's capability extends to more complex scenarios, such as use case II, where the learned loss for a four-service scenario is depicted in Fig. 7. This figure presents three specific perspectives of the complete eight-dimensional loss (two dimensions per client), taking into account anticipatory decisions and actual demand. This visualization underscores the impracticality of manually designing such a complex loss; yet, with Loss2Net, it is both achievable and interpretable.

Another key aspect of Loss2Net is its adaptability to fluctuating demands. The model is designed to learn a loss that adjusts for varying precision levels in predicting demands of different scales.

For use case III, the two plots in Fig. 8 show the loss and error distributions for the 10% of samples with lowest and highest traffic demands, under (a) the surrogate $\alpha$-OMC loss and (b) Loss2Net. While the learned loss in Loss2Net always captures an invariant general behavior, we can observe slight shifts in the error $\hat{y}_{t+1} - y_{t+1}$ that minimizes the cost, depending on the absolute value of $y_{t+1}$. In other words, Loss2Net learns a loss that naturally compensates for the different accuracy of the predictor in anticipating traffic values of diverse magnitude. Such an adaptation results in a significantly better prediction of low-volume demands, as shown by the major shift towards the origin of the green error distribution, when compared with the surrogate loss. This type of adjustments in the loss function are impossible to ascertain by just looking at the performance metric, i.e., the so-called *teacher*, as they inherently depend on the prediction quality. Yet, co-training the predictor and loss shaper block as done by our proposed model enables discovering a loss that is optimized for different absolute values of the predicted variable.

