# OpenReview forum: "Loss2Net: Loss Meta-Learning for Regression with A-priori Unknown Metrics"
_ICLR.cc/2025/Conference — ICLR 2025 Conference Withdrawn Submission_

### Official Review · Reviewer_kGaB · 2024-10-19

**Soundness:** 2
**Presentation:** 2
**Contribution:** 2
**Rating:** 3
**Confidence:** 5

**Summary:**

In this paper, the authors propose a method called Loss2Net, which aims to address the problem where the unknown prediction metric relationship is time-correlated, non-differentiable, or depends on intertwined predictions. The proposed method introduces a novel architecture and optimization approach that simultaneously learns the appropriate regressor and the loss function directly from system responses. Experiments on power grid and telecommunications infrastructure optimization demonstrate its effectiveness in real-world applications.

**Strengths:**

• The proposed method does not require prior knowledge of the (base) loss function structure as it is represented as a learnable meta-network, making it highly flexible for various application domains (despite the authors only really showing results in the domain of regression/time series).

• The loss function is leaned jointly with the regressor in the same backpropagation process by minimizing the L2 loss between the meta-learned loss and the target performance metric, which can help improve alignment with the target performance metric. The method is capable of approximating non-differentiable performance metrics, which is a desirable characteristic.

**Weaknesses:**

•	The proposed method optimizes the loss network by minimizing the L2 loss between the meta-learned loss and the target performance metric, both on the training set. The authors don’t discuss in detail how this is different from prior methods such as ML^3 which used unrolled differentiation, on the validation loss. In my opinion, it is not at all clear why the Loss2Net objective can improve testing performance. It would be beneficial to outline theoretically why this is the case and empirically compare performance against unrolled differentiation.

• In the background, the authors discuss multiple existing loss function learning methods, e.g., TaylorGLO, ML^3, EvoMAL. Despite this, there are no experiments directly comparing their proposed method to them, with only a modest number of experiments comparing to simple handcrafted loss functions such as MSE, MSLE, and $\alpha$-OMC

• The overall communication of the writing could be improved in my opinion, especially given that the proposed algorithm is relatively straightforward (Algorithm 1, page 6).

• The background discussion, lines 138 – 143, is not correct. You mentioned that methods such as GLO can’t be applied to regression, this is not correct. Furthermore, EvoMAL’s experiments are predominantly on single-task learning, not multi-task learning. Furthermore, given that these methods can also optimize for non-differentiable performance metrics, it is of high importance that the authors further highlight the novelty of their proposed method.

• I think for a conference such as ICLR which is less application-orientated relative to conferences such as KDD etc., I would have preferred more standard benchmarks that way readers could better gauge the performance of the proposed algorithm. For example, given the generality of the proposed method I don’t see why experiments couldn’t have been performed on CIFAR-10 or ImageNet, as opposed to power grid and telecommunications infrastructure optimization problems.

**Questions:**

See weaknesses

---

### Official Review · Reviewer_jTD7 · 2024-10-28

**Soundness:** 2
**Presentation:** 2
**Contribution:** 2
**Rating:** 5
**Confidence:** 4

**Summary:**

The authors of this submission aim to improve the supervision provided by the task-agnostic loss function for regression tasks by learning loss functions. Instead of hand-designing loss function for specific regression tasks, a loss function parameterised by a neural network approximating the target loss function to enable differentiable feedback signal is designed and learned during the task model training. The method is empirically tested on two datasets and demonstrates improvements compared to baseline losses.

**Strengths:**

1. The motivation is clear and practical.
2. The paper is clear and easy to follow.

**Weaknesses:**

1. The proposed algorithm is closer to learning a loss function approximator when the target signal is not differentiable than the meta-learning loss function which is commonly instantiated by learning a reusable loss function or an auxiliary loss to improve the model’s performance. In addition, the algorithm framework is not close to any existing meta-learning algorithm framework.

2. The experiments are very limited, using only shallow neural networks on two datasets. More applications and tasks can be found in [1]. The majority of the regression tasks in [1] are differentiable, so this can provide some task cases for whether the proposed algorithm can learn beyond the teacher.

3. The writing of the paper is not well-structured. Discussion and compassion with other methods are presented too much in the method section.

4. In remark 2, the prediction \hat{y} is fed into predictor DNN with random vectors and is claimed as novel by the authors. However, it has been commonly applied to improve the robustness of the model.

5. In the nondifferentiable cases, some gradient estimators, such as Reinforce, REBAR [2] and other advanced methods,  could be the baselines for comparison.

[1] Lathuilière S, Mesejo P, Alameda-Pineda X, Horaud R. A comprehensive analysis of deep regression. IEEE transactions on pattern analysis and machine intelligence. 2019 Apr 11;42(9):2065-81.

[2] Tucker G, Mnih A, Maddison CJ, Lawson J, Sohl-Dickstein J. Rebar: Low-variance, unbiased gradient estimates for discrete latent variable models. Advances in Neural Information Processing Systems. 2017;30.

**Questions:**

1. The loss landscape is highly discontinuous and illustrates multiple stages each each has a very flat curvature. Can the author explain more about how the model can be trained in this scenario? And I suppose that once the loss function landed in the discontinuous points some numerical problem such as Nan may happen.

2. Since the loss function is learned jointly with the main task model, the supervision of the loss function could be unstable while training. How does the learning curve of the model behave?

3. How the loss function is initialised?

---

### Official Review · Reviewer_K3VU · 2024-10-29

**Soundness:** 3
**Presentation:** 3
**Contribution:** 2
**Rating:** 5
**Confidence:** 3

**Summary:**

The paper introduces a novel framework, Loss2Net, designed for regression tasks where the optimal loss function is unknown a priori. Traditional loss functions like MSE may not align with task-specific performance metrics, leading to suboptimal predictions. Loss2Net addresses this by jointly training a predictor network and a "loss shaper" network, which learns the unknown loss function directly from system feedback, without prior assumptions.

**Strengths:**

This paper proposed an innovative method which handle the regression problem when the loss function is unknow, and this problem has not been studied previously. Provide an interested method that modeling the loss with a model (Loss2Net).

**Weaknesses:**

**Limited Theoretical Insight**: While the paper demonstrates empirical success, it lacks a rigorous theoretical foundation explaining why the proposed method performs well. Providing theoretical analysis on convergence, stability, and generalization properties would enhance the scientific rigor and credibility of Loss2Net. Such insights could help validate the model's robustness and its applicability across different problem settings, offering a deeper understanding of when and why this approach succeeds or encounters limitations.

**Lack of Synthetic Experiments on Benchmark Datasets**: Although Loss2Net shows superior performance over standard losses like MSE and MSLE, the reasons behind this improvement remain unclear due to limited experimental analysis. Including synthetic experiments on widely-used benchmark regression datasets, would better illustrate the conditions under which Loss2Net outperforms conventional methods. Or even the synthetic dataset would provide a better understanding on the method. An experiment on synthetic dataset can help better understanding how the loss shaper works. Instead of a enxperiment on the real-world datasets. For example generate different target metrics (loss functions) and see how your Loss2Net fit those target loss functions better than the simple loss function, and how those simple loss function lose efficacy.

Besides, using a benchmark experiments would also facilitate easier comparison for future research, enabling a clearer demonstration of Loss2Net’s advantages and limitations in standardized settings.

**Questions:**

The procedure of the hyperparameter tuning of the experiment was not provided in the papar. Please provide this part of details.

---

### Official Review · Reviewer_cDVC · 2024-11-02

**Soundness:** 2
**Presentation:** 2
**Contribution:** 2
**Rating:** 3
**Confidence:** 5

**Summary:**

This paper leverages loss meta-learning to address the issue of loss-metric mismatch by proposing a Loss2Net model designed to handle common regression tasks in real-world systems. The model is capable of jointly learning both the actual regressor and the loss function without any assumptions about the structure of the loss function. Experiments were conducted on several real-world cases, demonstrating that the model can learn previously unidentified loss functions.

**Strengths:**

- This paper proposes the Loss2Net model, in which a predictor and a loss shaper are designed separately. The predictor consists of a set of individual regressors, and the loss shaper is designed with an INR structure, which can learn geometric multidimensional representations based on coordinates.
- The Loss2Net model proposed in this paper does not require assumptions about the specific structure of the loss function, making it capable of handling real-world applications where the relationship between predictive variables and performance metrics is unknown in advance.
- Experiments were conducted on multiple real-world applications to demonstrate the advantages of Loss2Net.

**Weaknesses:**

- The motivation of this paper is not clearly explained. In the introduction, the issues with loss functions like MSE in real-world applications are described only in text, without any visual representation to provide a clearer illustration. This makes it unfriendly for researchers outside the engineering field and does not effectively convey the motivation of the paper.
- In Section 3.1, the paper alternately or iteratively optimizes the loss shaper and regressor through (2) and (3), but these two formulas do not form a bi-level optimization problem because the inner-layer regressor parameter θ^pdoes not depend on the outer-layer parameter $θ^l$. Therefore, it cannot be considered as meta-learning.
- In Section 3.2 of the paper, the authors only explain in text how the regressor and loss shaper are designed, and their contributions are mixed with existing research, failing to clearly highlight their own contributions.
- In the experimental section of the paper, the authors only describe the definitions of the two real-world application metrics in text but do not provide the mathematical symbols and meanings for each variable, especially how the metric M is calculated during execution. This is crucial for determining whether the optimization model's equation (2) can be computed, but the paper does not address how $M=f_M (y ̂,v)$ is derived in the experiments.

**Questions:**

-How can the real performance metric M be derived from y ̂ and v to calculate equation (2) in your experiments?
-The definition of the metric is not clearly explained, and the variables in the datasets are not sufficiently stated to match the notation in the paper.
-The paper lacks several key visualizations that would better illustrate the motivations behind the work, making it harder for researchers outside the engineering field to fully grasp its advantages. To enhance readability, please consider using more symbols and visual aids where appropriate.

---

### Official Review · Reviewer_cuVx · 2024-11-05

**Soundness:** 3
**Presentation:** 3
**Contribution:** 2
**Rating:** 6
**Confidence:** 3

**Summary:**

The paper presents Loss2Net, a meta-learning framework for regression tasks with unknown performance metrics, where traditional loss functions (like MSE) may be ineffective. Loss2Net jointly trains a predictor and a loss-shaper network, enabling it to learn both optimal predictions and the corresponding loss function directly from system feedback. This approach accommodates complex, non-differentiable, and time-correlated loss functions. Through experiments in the power grid and telecom resource management, Loss2Net shows improved adaptability over standard methods, demonstrating its potential for real-world engineering applications.

**Strengths:**

1. The paper studies an important problem to use meta learning in real-world regression problems with unknown loss functions.

2. The paper proposes a method to learn non-differentiable, multi-dimensional losses without prior assumptions.

**Weaknesses:**

1. The proposed method relies on high-quality system response data, which may limit robustness with noisy data.

2. The dual-component architecture and the joint optimization may result in higher computational requirements and complexity, making it less suitable for lightweight or real-time applications.

**Questions:**

1. The proposed method relies on high-quality system response data, how does it perform when the system response data contain noise?

---

### Note · Authors · 2024-11-19

**Comment:**

We are very grateful to the reviewers for their constructive comments. This comments will help us to improve both the presentation and the completeness of the proposed concept.

We consider that the received comments require a substantial revision of the manuscript, and we plan to perform new experiments to validate the generality of the solution. Therefore, we would like to withdraw the submission to work on improving the work.

We would like to thank again the Reviewers and the Chairs for the time invested in their insightful review of our submission.

**Withdrawal Confirmation:**

I have read and agree with the venue's withdrawal policy on behalf of myself and my co-authors.